# Association between human herpesvirus infections and dementia or mild cognitive impairment: a systematic review protocol

Charlotte Warren-Gash,[1] Harriet Forbes,[1] Judith Breuer,[2] Andrew C Hayward,[3] Angelique Mavrodaris,[4] Basil H Ridha,[5] Martin Rossor,[5,6] Sara L Thomas,[1] Liam Smeeth[1]

► Prepublication history and additional material are available. To view these files please visit the journal online (http://dx.doi.org/10.1136/bmjopen-2017-016522).

¹Faculty of Epidemiology and Population Health, London School of Hygiene and Tropical Medicine, London, UK
²Division of Infection and Immunity, University College London, London, UK
³Institute of Epidemiology and Health Care, University College London, London, UK
⁴Cambridge Institute of Public Health, Cambridge, Cambridgeshire, UK
⁵NIHR Queen Square Dementia Biomedical ResearchUnit, University College London, London, UK
⁶Dementia Research Centre, Institute of Neurology, University College London, Queen Square, London, UK

**Correspondence to**
Dr Charlotte Warren-Gash; Charlotte.Warren-Gash1@lshtm.ac.uk

## ABSTRACT

**Introduction** Persisting neurotropic viruses are proposed to increase the risk of dementia, but evidence of association from robust, adequately powered population studies is lacking. This is essential to inform clinical trials of targeted preventive interventions.

**Methods and analysis** We will carry out a comprehensive systematic review of published and grey literature of the association between infection with, reactivation of, vaccination against or treatment of any of the eight human herpesviruses and dementia or mild cognitive impairment. We will search the Cochrane Library, Embase, Global Health, Medline, PsycINFO, Scopus, Web of Science, clinical trials registers, the New York Academy of Medicine Grey Literature Report, Electronic Theses Online Service through the British Library and the ISI Conference Proceedings Citation Index for randomised controlled trials, cohort, case–control, case crossover or self-controlled case series studies reported in any language up to January 2017. Titles, abstracts and full-text screening will be conducted by two researchers independently. Data will be extracted systematically from eligible studies using a piloted template. We will assess risk of bias of individual studies in line with the Cochrane Collaboration tool. We will conduct a narrative synthesis, grouping studies by exposure and outcome definitions, and will describe any differences by population subgroups and dementia subtypes. We will consider performing meta-analyses if there are adequate numbers of sufficiently homogeneous studies. The overall quality of cumulative evidence will be assessed using selected Grading of Recommendations, Assessment, Development and Evaluations criteria.

**Ethics and dissemination** As this is a review of existing studies, no ethical approval is required. Results will be disseminated through a peer-reviewed publication and at national and international conferences. We anticipate the review will clarify the current extent and quality of evidence for a link between herpesviruses and dementia, identify gaps and inform the direction of future research.

**Prospero registration number** CRD42017054684.

### Strengths and limitations of this study

► This systematic review will comprehensively evaluate published and grey literature on the association between infection with, reactivation of, vaccination against or treatment of any of the eight human herpesviruses and dementia or mild cognitive impairment.

► One key strength of our review is that it will assess the evidence for differential effects of human herpesviruses on dementia risk in different population subgroups, for example, by age, ethnicity and immune status, to inform the design and targeting of randomised controlled trials of preventive interventions to reduce dementia risk.

► A lack of adequately powered studies or heterogeneity of studies selected for inclusion in terms of study design, exposure and outcome definitions or study population may restrict our capacity to derive clear conclusions about the study question.

## INTRODUCTION
### Rationale

Dementia has now overtaken ischaemic heart disease as the leading cause of death in England and Wales.[1] The prevalence is forecast to rise rapidly as the population ages, with the number of dementia cases worldwide projected to reach 131.5 million in 2050 compared with 46.8 million in 2015,[2] assuming that age-specific prevalence rates remain stable.[3] After accounting for overlap between risk factors, only around one-third of dementia cases are attributable to potentially preventable factors,[4] and there are no effective treatments. It is therefore essential to identify other modifiable determinants.

Viral aetiologies for dementia have been postulated for decades, but population-level evidence is inconsistent and no antiviral strategies to reduce dementia risk are in routine clinical use. Recent interest has focused on

herpes simplex virus type 1 (HSV1),[5 6] one of eight human herpesviruses that routinely infect humans. After primary infection, all human herpesviruses establish latency within specific tissues, with the alpha herpesviruses HSV1, HSV2 and varicella zoster virus (VZV) demonstrating neurotropism through their characteristic persistence in sensory nerve ganglia.[7]

HSV1 infection of neuronal and glial cell cultures induces cellular changes similar to those seen in Alzheimer's disease (AD),[8–10] and which are reversible using antiviral agents.[11] Postmortem studies show that HSV establishes latency in human brain tissue: viral genomic sequences have been detected in up to 35% of neurologically asymptomatic individuals[12]; HSV reactivation from latency occurs in explanted mouse brainstems.[13 14] A meta-analysis of mainly small case–control studies of varying quality published in 2015 suggested that HSV1 seropositivity was associated with an increased risk of AD (pooled OR 1.38 (95% CI 1.03 to 1.84)) as was the presence of Epstein-Barr virus (EBV) antibodies (pooled OR 1.55 (95% CI 1.12 to 2.13)).[15] Although the presence of any of six herpesviruses in brain tissue was associated with AD in case–control studies (pooled OR 1.38 (95% CI 1.14 to 1.66)), the temporal sequence and clinical significance of this finding are unclear.

Our study will have some key differences to this previous review: we will consider the effect of infection with and reactivation of all eight human herpesviruses on both AD and other neurodegenerative diseases causing dementia or mild cognitive impairment (MCI); we will investigate whether preventing or treating herpesvirus infections affects dementia risk; and we will assess the evidence for differential effects of human herpesviruses on dementia risk in different population subgroups, for example, by age, ethnicity, socioeconomic status and immune status, to inform the design and targeting of randomised controlled trials (RCTs) of preventive interventions.

## Objectives
The overall aim is to determine whether people infected with any of the eight human herpesviruses are at greater risk of subsequently developing dementia or MCI than those without evidence of herpesvirus infection.

Specific research questions include the following:
1. Does primary infection with or reactivation of human herpesviruses affect the risk of dementia or MCI?
2. Does preventing or treating human herpesviruses modulate dementia or MCI risk?
3. Does any association between human herpesviruses and dementia or MCI vary by population subgroups or between subtypes of dementia?

## METHODS
This systematic review protocol has been prepared according to the Preferred Reporting Items for Systematic Review and Meta-Analysis Protocols 2015 statement.[16]

## Eligibility criteria
### Study design
We will include all RCTs, prospective and retrospective cohort studies, case–control studies, self-controlled case series and case crossover studies that either present an estimate of effect or provide sufficient data for an effect estimate to be calculated.

### Population
Our study population will include adults aged 18 years and over, with results stratified where possible by age group, immune status and apolipoprotein E4 (APOE4) status. Studies conducted in any setting, for example, hospital inpatient, outpatient, primary care or the community, will be considered. There will be no language or geographical limits. No animal studies will be included.

### Exposure
Our exposures are infection with or reactivation of any of the eight human herpesvirus infections (defined either clinically or through appropriate laboratory criteria), herpesvirus vaccinations (eg, Zostavax) and herpesvirus treatments (eg, with antiviral agents such as acyclovir).

### Comparators
Comparators will vary by study design and will include subjects randomised not to receive an antiviral intervention or vaccination (RCTs), people unexposed to herpesvirus infections (cohort studies and case–control studies) and person time unexposed to herpesviruses (self-controlled case series or case crossover studies).

### Outcome
The primary outcomes are dementia (all types), diagnosed either clinically, with or without neuroimaging or by histopathology, and MCI, characterised clinically. Where possible, dementia will be further categorised by type, for example, AD, vascular dementia, frontotemporal dementia and its subcategories, Lewy body dementia, and other rare subtypes including Huntington's disease and prion diseases.

## Literature searches
We will search the Cochrane Library, Embase, Global Health, Medline, PsycINFO, Scopus and Web of Science from dates of inception to January 2017. We will also search clinical trials registers and grey literature, including ClinicalTrials.gov, the New York Academy of Medicine Grey Literature Report (www.greylit.org), the Electronic Theses Online Service through the British Library (http://ethos.bl.uk) and the ISI Conference Proceedings Citation Index (http://isknowledge.com). We will use medical subject heading (MeSH) terms and keyword searches (in the title and abstract) to capture studies of infection with, vaccination against or antiviral treatment of, any of the eight human herpesviruses (HSV1, HSV2, VZV, EBV, cytomegalovirus (CMV), human herpesvirus 6 (HHV6), HHV7, HHV8) and dementia (including subtypes) or MCI. This strategy will be supplemented

by reviewing reference lists of eligible articles and relevant reviews. The search strategy has been developed in Medline and will be translated for use in other databases (see online supplementary appendix 1).

## Study records
### Data management
Search results will be uploaded into an EndNote database (V.7.5) and de-duplicated.

### Selection process
All titles and abstracts will be scanned for eligibility by two researchers in parallel. The full texts of all articles that potentially meet the inclusion criteria will then be obtained and again reviewed in parallel. We will note reasons for rejection of articles at this stage according to a hierarchical list (ineligible study design, wrong exposure, wrong outcome, insufficient information to calculate an effect estimate). Any discrepancies will be discussed between reviewers, and consultation with a third reviewer will be carried out where necessary.

### Data collection process
The data items listed below will be incorporated into a pilot data extraction table. Parallel data extraction will be carried out for the first three included studies by two reviewers and changes to the extraction table made as required. Any discrepancies will be discussed and resolved through consultation with a third reviewer if necessary. Data will be extracted for each remaining study by a single reviewer. We will also consider contacting corresponding authors of published studies to obtain any further information needed using a standardised email template.

### Data items
We will design a data extraction table to collect information on the following domains:
► Population: characteristics of the study population (eg, sex and age distribution, ethnicity, immune status, APOE4 status), recruitment and sampling methods, inclusion/exclusion criteria;
► Exposure: exposure status definition and identification, number of exposed subjects, any exclusions;
► Comparators: identification and definition of unexposed individuals, number of unexposed subjects, any exclusions;
► Outcomes: definition and identification of primary (dementia or MCI) and secondary outcomes (dementia subtypes), number of subjects, any exclusions;
► Study characteristics: authors, publication year, setting/source of participants, design, period of study, length of follow-up time (if relevant), aims and objectives.

We will record both unadjusted and fully adjusted effect estimates for the association between each of the human herpesviruses and risk or rate of dementia or MCI.

Details of the confounders measured and adjusted for will be noted. Results of any additional stratified analyses will also be recorded, for example, on virus effects in different population subgroups or on dementia subtypes, including categories of severity.

## Outcomes and prioritisation
The primary outcomes are dementia (all types) and MCI. Where possible, the neurodegenerative condition leading to the syndromes of dementia or MCI will be recorded, for example, AD, vascular dementia, mixed AD and vascular dementia, frontotemporal dementia and its subcategories, Lewy body dementia, and other rare subtypes including Huntington's disease and prion diseases. A priori, we anticipate that most studies will report on AD. We have, however, opted for a broader primary outcome definition: excluding studies that do not record dementia type will limit the statistical power to detect an effect, yet identifying a precise dementia aetiology is challenging, especially when histopathological diagnosis is not available. We will prioritise studies in which herpesvirus status was ascertained prior to the occurrence of an outcome event when assessing study quality. These are likely to be longitudinal prospective studies that generate incidence rate ratios or HRs, but we will also consider estimates of ORs from case–control studies.

## Risk of bias in individual studies
We will assess the risk of bias in individual studies in line with the Cochrane Collaboration approach for both randomised and non-randomised studies.[17 18] For RCTs this will include consideration of the effects of selection bias, performance bias, detection bias, attrition bias and reporting bias. For observational studies we will consider relevant domains, including selection of participants, measurement of variables, control for confounding and missing data. For each study, each component will be assigned a risk of bias category 'high risk', 'low risk' or 'unclear risk', and a summary risk of bias table produced. Two reviewers will assign risk of bias categories in parallel for the first three studies and, as before, any discrepancies that cannot be resolved will be discussed with a third reviewer. One reviewer will then assign risk of bias categories for the remaining studies.

## Data synthesis
We will conduct a narrative synthesis in which evidence is grouped by exposure and outcome definitions, and we will additionally summarise the results in tables. We will assess whether there are sufficient data to investigate the effect of herpesviruses on population subgroups or subtypes of dementia. We will investigate whether vaccination to prevent herpesvirus infection or reactivation, for example, using chickenpox vaccine or herpes zoster vaccine mitigates the risk of dementia. We will also present any evidence for a modulating effect of antiviral treatment against herpesviruses on dementia.

We will consider performing a meta-analysis if there are sufficient numbers of studies with the same design and sufficiently homogeneous populations, exposures and outcomes to calculate pooled effect estimates. For case–control studies we will present pooled ORs, for cohort studies, pooled rate ratios or HRs, and for case-only designs, pooled incidence ratios will be calculated. The choice of fixed or random effects models will be guided by the level of statistical heterogeneity. We will consider the magnitude and direction of effects when interpreting $I^2$ values, but will generally take an $I^2$ value of >25% to represent moderate or substantial heterogeneity.[19 20] Sources of heterogeneity will be explored by considering the use of meta-regression to compare summary estimates from different study-level characteristics, such as age of the study population, study design and dementia or MCI diagnosis. In sensitivity analysis, the most biased studies (those with more than one domain classified as high risk of bias) will be excluded. If there are sufficient numbers of studies, we will investigate the risk of publication bias using a funnel plot. All of the statistical analysis will be performed using STATA V.14.0.

## Cumulative evidence

The quality of cumulative evidence across studies for an association of each herpesvirus with each outcome will then be assessed using selected Grading of Recommendations, Assessment, Development and Evaluations criteria that are relevant to the included studies.[21] As well as risk of bias, these include inconsistency, indirectness, imprecision, publication bias and any additional domains deemed appropriate to categorise the strength of evidence as 'high', 'moderate' or 'low/very low'.

## Ethics and dissemination

Ethical approval was not required as this study is a systematic review of previously published studies. The protocol was registered with the International Prospective Register of Systematic Reviews (PROSPERO) on 7 January 2017 (Registration number: CRD42017054684). Any future amendments will be documented on the PROSPERO website. Results will be submitted for publication in a peer-reviewed journal and presented at national and international conferences.

**Contributors** CWG conceived and designed the study, drafted and revised the protocol following author comments; HF contributed to the design of the study and revised the paper critically; JB contributed to the conception and design of the study and revised the paper critically; ACH contributed to the conception and design of the study and revised the paper critically; AM contributed to the design of the study and revised the paper critically; BHR contributed to the design of the study and revised the paper critically; MR contributed to the design of the study and revised the paper critically; SLT contributed to the design of the study and revised the paper critically; LS contributed to the conception and design of the study and revised the paper critically. All authors approved the final version of the protocol.

**Competing interests** None declared.

**Patient consent** Not applicable: systematic review protocol.

**Provenance and peer review** Not commissioned; externally peer reviewed.

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
