## [Reviewer comments · BMJ Open]

ARTICLE DETAILS

TITLE (PROVISIONAL)	Association between human herpesvirus infections and dementia or mild cognitive impairment: a systematic review protocol
AUTHORS	Warren-Gash, Charlotte; Forbes, Harriet; Breuer, Judith; Hayward, Andrew; Mavrodaris, Angelique; Ridha, Basil; Rossor, Martin; Thomas, Sara; Smeeth, Liam

VERSION 1 - REVIEW

REVIEWER	Timo Strandberg University of Helsinki and Helsinki University Hospital, Helsinki, Finland; University of Oulu, Center for Life Course health Research, Oulu, Finland
REVIEW RETURNED	09-Mar-2017

GENERAL COMMENTS	It is important to make this systematic review and it adds to previous papers on this topic. I don't have remarks on the general, careful outline of the analysis. My specific comments: 1. Dementia is a problematic term and should be reserved to the end-stage of cognitive disorders. However, it is still commonly used.2. Introduction, 1st para: One third of dementia cases... preventable. Also higher proportions (50 to 60%) have been presented. Please check the newest reviews (eg. Lancet Neurology) on this.3. This recent review should be referenced in Introduction, for example in 2nd para. Itzhaki RF1, Lathe R2, Balin BJ et al. Microbes and Alzheimer's Disease. J Alzheimers Dis. 2016;51(4):979-84. doi: 10.3233/JAD-160152. However, I agree with the authors that this systematic review can focus on herpesviruses.4. Outcome and prioritization: This is quite complicated because diagnosis in old age is usually clinical in studies and not based on neuropathological examination. Not all "Alzheimer" is "pure" Alzheimer, but viruses may be related specifically to pure Alzheimer. Please try to specify also mixed (AD+vascular) and differentiate it from AD without vascular features. This is probably difficult, though.5. APOE: please prespecify APOE4 vs non APOE4, if reported in studies; virus etiology may be related to APOE4.
--

REVIEWER	JIANMIN XING Beijing TIANXIE Medical & Technology Ltd. Co. China
REVIEW RETURNED	20-Apr-2017

GENERAL COMMENTS	Thanks for your efforts and hard work on this topic. Some suggestions: 1. "We will assess the risk of bias in individual studies in line with the Cochrane Collaboration approach¹⁵." Here, This is only for RCT. You should provide exclusive methods/tools they will use to assess the risk of bias for cohort studies, case control studies, case crossover studies and case series. As long as I know, all those different types of studies use different tools. You should provide reference for each of them. 2. In the section of "Data Synthesis", The statistical methods that could be used to pool data from case-crossover studies and case series, eg. one sample rate. You should provide exclusive methods for analyzing different types of studies. And the data comes from case control studies should be handled differently from prospective studies which group the participants according to intervention/exposure. You should present more details of the handling of data from case-control studies. 3. In Line 55 Page 9 of 15, how to define the "homogeneous" ? And in Line 1 Page 10 of 15, Please give the cutting point of I-square for deciding random or fixed effect models. 4 Please provide reason why the meta-regression is planned to be done after removal of high risk of bias trials. And, in the methods section, there is no definition on how to judge a trial is of high/low risk of bias as a whole (not separate items).
--

VERSION 1 – AUTHOR RESPONSE

Reviewer: 1

It is important to make this systematic review and it adds to previous papers on this topic. I don't have remarks on the general, careful outline of the analysis. My specific comments:

1. Dementia is a problematic term and should be reserved to the end-stage of cognitive disorders. However, it is still commonly used.

Response: We agree that 'dementia' is a description of late stage cognitive impairment arising from different underlying neurodegenerative processes rather than a diagnosis in itself. Nevertheless, 'dementia' is a commonly used term understood by health policymakers, clinicians outside the field and the lay public. We want to ensure that this review is easily identifiable to any groups seeking evidence to inform dementia research, prevention and care. In 'Outcomes and prioritization' we now clarify that we aim to describe effects of herpesviruses on each neurodegenerative condition leading to the syndromes of dementia or mild cognitive impairment. We also add mild cognitive impairment to the protocol title.

2. Introduction, 1st para: One third of dementia cases... preventable. Also higher proportions (50 to 60%) have been presented. Please check the newest reviews (eg. Lancet Neurology) on this.

Response: We agree that estimates of the proportion of potentially preventable dementia cases range from 33% to 60% across different studies. One limitation of the higher estimates of 50 to 60% is that they do not take into account non- independence between modifiable risk factors. We now make the

reason for our choice of estimate clear in the introduction on p4.

3. This recent review should be referenced in Introduction, for example in 2nd para. Itzhaki RF1, Lathe R2, Balin BJ et al. Microbes and Alzheimer's Disease. J Alzheimers Dis. 2016;51(4):979-84. doi: 10.3233/JAD-160152. However, I agree with the authors that this systematic review can focus on herpesviruses.

Response: We thank the reviewer for highlighting this paper, which is now referenced on p4.

4. Outcome and prioritization: This is quite complicated because diagnosis in old age is usually clinical in studies and not based on neuropathological examination. Not all "Alzheimer" is "pure" Alzheimer, but viruses may be related specifically to pure Alzheimer. Please try to specify also mixed (AD+vascular) and differentiate it from AD without vascular features. This is probably difficult, though.

Response: Although we agree that the ability to classify the underlying pathological process leading to dementia is likely to be limited, we have added the category 'mixed' to our secondary outcome (dementia subtype) to capture any cases of mixed AD and vascular pathology (p8).

5. APOE: please prespecify APOE4 vs non APOE4, if reported in studies; virus etiology may be related to APOE4.

Response: Under 'eligibility criteria' and 'data extraction', we now specify that we will extract and present data on APOE4 status, where available.

Reviewer: 2

Thanks for your efforts and hard work on this topic. Some suggestions:

1. "We will assess the risk of bias in individual studies in line with the Cochrane Collaboration approach¹⁵." Here, This is only for RCT. You should provide exclusive methods/tools they will use to assess the risk of bias for cohort studies, case control studies, case crossover studies and case series. As long as I know, all those different types of studies use different tools. You should provide reference for each of them.

Response: We now reference the Cochrane collaboration tool for assessing risk of bias for non-randomised studies (ROBINS-I), which builds on previous approaches for assessing quality of non-randomised studies such as the Newcastle-Ottawa scale. Although ROBINS-I was originally designed for non-randomised studies of interventions, the tool assesses risk of bias due to confounding, selection of participants, misclassification of exposure, missing data, measurement of outcomes and selective reporting that apply to all of our included observational study designs. Where possible we aim to harmonise the approach to assessing risk of bias across all studies rather than using separate tools for study type, and an approach based on the ROBINS-I tool does enable assessment of risk of bias for each type of observational study.

2. In the section of "Data Synthesis", The statistical methods that could be used to pool data from case-crossover studies and case series, eg. one sample rate. You should provide exclusive methods for analyzing different types of studies. And the data comes from case control studies should be handled differently from prospective studies which group the participants according to intervention/exposure. You should present more details of the handling of data from case-control

studies.

Response: We have now added a section on pooling data for each of our included study designs, clarifying that we will handle data from case control, cohort and case only studies separately (p9-10). As the study aims to look at the association between herpes viruses and dementia onset, we will restrict to studies with a comparison group or comparison time period (i.e. case series that simply measure 'rates' will be excluded).

3. In Line 55 Page 9 of 15, how to define the "homogeneous" ? And in Line 1 Page 10 of 15, Please give the cutting point of I-square for deciding random or fixed effect models.

Response: We have now reworded the section on meta-analysis to describe further the situations in which we will consider conducting a meta-analysis. We will also consider an I² value of 25% or above to represent moderate or substantial heterogeneity and guide the use of a random effects model(1), while recognising that the interpretation of I² values also depends on the magnitude and direction of effects.

4. Please provide reason why the meta-regression is planned to be done after removal of high risk of bias trials. And, in the methods section, there is no definition on how to judge a trial is of high/low risk of bias as a whole (not separate items).

Response: We now clarify that meta-regression of all eligible studies is planned, but in a separate sensitivity analysis we will investigate the effect of removing studies with components that suggest a high risk of bias on pooled effect estimates. We do not plan to give individual studies an overall rating for risk of bias but rather will assess specific domains separately (following the recommended Cochrane approach) and present these in a summary table. For this sensitivity analysis, the most biased studies (those with more than one domain classified as high risk of bias) will be excluded. We will then consider the strength of cumulative evidence for the effect of each herpesvirus exposure on dementia outcomes across studies using the GRADE approach. This is now clarified on p9-10.

References

1. Riley RD, Higgins JPT, Deeks JJ. Interpretation of random effects meta-analyses. *BMJ* 2011; 342:d549